# The impact of variance in carnitine palmitoyltransferase-1 expression on breast cancer prognosis is stratified by clinical and anthropometric factors

**Ryan Liu[1,2], Shyryn Ospanova[3], Rachel J. Perry [1]***

**1** Department of Internal Medicine and Cellular & Molecular Physiology, Yale School of Medicine, New Haven, Connecticut, United States of America, **2** Cedar Park High School, Cedar Park, Texas, United States of America, **3** University of Pennsylvania, Philadelphia, Pennsylvania, United States of America

\* rachel.perry@yale.edu

## Abstract

CPT1A is a rate-limiting enzyme in fatty acid oxidation and is upregulated in high-risk breast cancer. Obesity and menopausal status' relationship with breast cancer prognosis is well established, but its connection with fatty acid metabolism is not. We utilized RNA sequencing data in the Xena Functional Genomics Explorer, to explore *CPT1A*'s effect on breast cancer patients' survival probability. Using [$^{18}$F]-fluorothymidine positron emission tomography-computed tomography images from The Cancer Imaging Archive, we segmented these analyses by obesity and menopausal status. In 1214 patients, higher *CPT1A* expression is associated with lower breast cancer survivability. We confirmed a previously observed protective relationship between obesity and breast cancer in pre-menopausal patients and supported this data using two-sided Pearson correlations. Taken together, these analyses using open-access databases bolster the potential role of CPT1A-dependent fatty acid metabolism as a pathogenic factor in breast cancer.

## Introduction

Breast cancer is the leading cause of cancer mortality in women [1]. Over two million new cases of breast cancer are diagnosed each year, accounting for 24% of total new cancer cases and 15% of total cancer deaths in women. There are many pathogenic and permissive mechanisms underlying tumorogenesis, and metabolic reprogramming is among these emerging mechanisms. Metabolic reprogramming explores how metabolic shifts activate pathways generating and utilizing substrates such as fatty acids and glucose [2]. These pathways provide the energy required for cancer cells to proliferate, migrate, and invade. Although, since the pioneering work of Otto Warburg, the cancer metabolism field has primarily focused on glucose as the most important substrate for rapidly proliferating tumor cells, the role of fatty acids in breast cancer metabolism is far less well understood. It is of particular importance to define how fatty acid metabolism affects tumor metabolism and, ultimately, growth in tumors such as

**Data Availability Statement:** These are third-party data and not the property of the authors. All data are available through The Cancer Imaging Archive

and The Cancer Genome Atlas. The links to the specific datasets are as follows: https://xenabrowser.net/datapages/?dataset=TCGA.BRCA.sampleMap%2FBRCA_clinicalMatrix&host=https%3A%2F%2Ftcga.xenahubs.net&removeHub=https%3A%2F%2FXena.treehouse.gi.ucsc.edu%3A443 https://gtexportal.org/home/gene/CPT1A https://wiki.cancerimagingarchive.net/pages/viewpage.action?pageId=30671268.

**Funding:** This study was supported in part by a Pilot Grant from the Yale Cancer Center, and by a dkNET Summer of Data student fellowship. The funders had no role in study design, data collection and analysis, decision to publish, or preparation of the manuscript.

**Competing interests:** The authors have declared that no competing interests exist.

breast cancer that exist in an adipocyte-enriched milieu, because the surrounding lipids are expected to heavily influence the composition of the tumor microenvironment.

Fatty acids are a nutrient-dense source of energy for cancer cells, and under certain conditions may be required for tumor cell survival [3]. Relatedly, lipids are a critical component of cell membranes, so it is not surprising that fatty acid oxidation is an increasingly well-established factor promoting cancer metastasis [4]. Carnitine palmitoyl transferase 1A (CPT1A) resides in the outer mitochondrial membrane [5] and is the first rate-limiting enzyme in fatty acid oxidation, via its role in mediating fatty acid entry into the mitochondria [2]. Increased CPT1 expression and/or activity would therefore be expected to contribute to increased cancer proliferation and, in turn, mortality, but, to our knowledge, this hypothesis remains surprisingly unproven. Increasing availability of multi-Omics data highlights the opportunity for prognostic analyses such as these [6, 7].

Proliferation rate is a prognostic marker for breast cancer patients, and it can be measured by positron emission tomography-computed tomography (PET-CT) with [$^{18}$F]-fluorothymidine ($^{18}$F-FLT) in humans [8]. $^{18}$F-FLT is a substance uptaken by cells in the S-phase of the cell cycle. Increased proliferative activity is seen as increased $^{18}$F-FLT uptake in breast cancer since high $^{18}$F-FLT uptake is observed in tissues presenting high mitotic activity. Therefore, increased tumor $^{18}$F-FLT uptake is generally viewed as a poor prognostic marker. We demonstrated in an analysis of PET-CT images from The National Cancer Imaging Archive (TCIA) that body mass index (BMI) is paradoxically negatively correlated with $^{18}$F-FLT uptake in premenopausal breast cancer patients but positively correlated with $^{18}$F-FLT uptake in postmenopausal breast cancer patients, as confirmed in the literature [9]. BMI is a powerful metric for cancer research, and we considered data from patients across a comprehensive range of body size classes based on BMI: 1—cachexia (BMI below 18.5; although no patients in the current study qualified for this category), 2—healthy weight (BMI of 18.5 to 24.9), 3—overweight (BMI of 25.0 to 29.9), and 4—obese (BMI of or above 30.0) [10].

Given the relatively small sample size of $^{18}$F-FLT scans available in the TCIA dataset (33 total), RNA Sequencing Visualization was used to confirm the image analysis results [11]. We employed the Xena Functional Genomics Explorer to understand how *CPT1A* expression may predict breast cancer prognosis, through the lens of clinical and anthropometric data. This is a unique aspect of the current study, as the literature has not yet explored how fatty acid metabolism predicts outcomes in breast cancer in the context of obesity and menopausal status.

This study employed a multimodal approach to understand how fatty acid metabolism and obesity status affect tumor thymidine uptake and proliferation. We confirmed prior evidence of the paradoxical relationship between obesity and breast cancer and the detrimental nature of *CPT1A* expression on prognoses [1, 12]. In addition, we stratified patients with high and low expression of *CPT1A* and analyzed how menopausal status affected the survival rate of each, imparting a new perspective on an explored topic. These results add to our understanding of how body composition and fatty acids can alter the metabolism of breast cancer, highlighting a compelling need for future mechanistic studies.

## Materials and methods

### RNA sequencing visualization

RNA sequencing data from the TCGA Breast Cancer (BRCA) data set was uploaded into the UCSC Xena Functional Genomics Browser's Visualization suite. The dataset used can be found here: https://xenabrowser.net/datapages/?dataset=TCGA.BRCA.sampleMap%2FBRCA_clinicalMatrix&host=https%3A%2F%2Ftcga.xenahubs.net&removeHub=https%3A%2F%2FXena.treehouse.gi.ucsc.edu%3A443. All 1,247 samples were included in column A.

*CPT1A* was added into column B as a genomic data type, and gene expression and somatic mutation datasets were selected to be analyzed. A Kaplan Meier plot was generated from column B, with overall survival chosen as the dependent variable. 1214 of the 1247 total samples contained data on the gene expression level of *CPT1A*, and these samples were used to generate the KM plot. Two groups of gene expression levels were set, with 602 samples considered to have a low expression level of *CPT1A* ($<$10.83 FPKM) and 612 samples considered to have a high expression level of *CPT1A* ($>$ = 10.83 FPKM). 10.83 FPKM was the cutoff level for CPT1A gene expression, as the samples were divided at the median. Lastly, the custom survival time cutoff was set to 8605 days after the first treatment. A KM plot was then generated from column C, with overall survival again chosen as the dependent variable. 790 out of the 1247 total samples contained data on mutations in the *CPT1A* gene, and these samples were used to generate the KM plot. The custom survival time cutoff was set to 8605 days after the first treatment.

Menopausal status was added as a phenotypic data type in the same Visualization suite. Two Kaplan Meier plots were generated from the *CPT1A* gene expression and menopausal status data: one depicting the correlation between menopausal status and survival probability in samples with high expression levels of *CPT1A* ($>$ = 10.83 FPKM) and the correlation between menopausal status and survival probability in samples with low expression levels of *CPT1A* ($<$10.83 FPKM). A new subgroup column was created: gene expression levels greater than or equal to 10.83 FPKM from column B were picked as the samples of interest and were filtered to keep those samples. A KM plot was generated for column C, with overall survival chosen as the dependent variable. 612 of the 1247 samples matched the criteria of high expression level of *CPT1A* ($>$ = 10.83 FPKM), and 545 of the 612 samples contained data on menopausal status. 514 of the 545 samples had clear distinctions between pre-menopausal and post-menopausal, and were used to generate the KM plot. 123 samples had pre-menopausal status, and 391 samples had post-menopausal status. The survival time cutoff was set to 3461 days after the first treatment. The current sample filter was cleared, and a new subgroup column was created: gene expression levels less than 10.83 FPKM from column B were picked as the samples of interest and filtered to keep those samples. A KM plot was generated for column C, with progression during disease-free interval chosen as the dependent variable. 606 of the 1247 samples matched the criteria of low expression level of *CPT1A* ($<$10.83 FPKM), and 552 of the 606 samples contained data on menopausal status. 506 of the 552 samples had clear distinctions between pre-menopausal and post-menopausal, and were used to generate the KM plot. 136 samples had pre-menopausal status, and 370 samples had post-menopausal status. The survival time cutoff was set to 3663 days after the first treatment.

Using the Model organism Aggregated Resources for Rare Variant ExpLoration (MARR-VEL) database, *CPT1A* data in humans was entered and transferred into the Genotype-Tissue Expression (GTEx) Portal [13–15]. The dataset used can be found here: https://gtexportal.org/home/gene/CPT1A. From these data, *CPT1A* expression level in different tissue types was examined.

## PET-CT image analysis

The scans used for the image analysis were provided by TCIA. The dataset can be found here: https://wiki.cancerimagingarchive.net/pages/viewpage.action?pageId=30671268. All patients with an [18]F-FLT PET-CT scan, height, weight, and menopausal status were studied, and 18 pre-menopausal patients and 15 post-menopausal patients fulfilled these criteria. If multiple scans were available, the earliest one was used to minimize the effect of neoadjuvant chemotherapy on the tumor.

PET and CT images were uploaded into Fiji ImageJ with a PET-CT Viewer. Using the brown fat ROI tool, the region of interest was drawn and projected through a range of slices. Fixed-volume spheres were drawn to measure mean thymidine uptake in tumor tissue, where only SUV parameters (2–15) and the "any" voxel criteria were selected. Lean body mass-corrected standardized uptake values (SUV) were calculated. The primary endpoint was SUV (g/mL) of tumors correlated to obesity status. Among 33 patients, BMI ranged from 19.0 to 33.3 kg/m$^2$ (mean [SD] = 27.2 [4.00]).

Data were analyzed using GraphPad Prism, and a two-sided Pearson correlation was performed. It was found that none of the analyses were normally distributed. Under this condition, a Mann-Whitney Test was performed on each analysis to determine significance. Statistical significance was defined as P values less than .05. Marginally significant values were determined as P values greater than .05 but less than .10, as none of the analyses produced significant results.

## Results

### Survival probability By *CPT1A* expression level and presence of mutations

Patients with a high expression level of *CPT1A* had a lower survival rate (median survival time of 3,500 days) than those with low expression levels of *CPT1A* (median survival time of 4,200 days) (Fig 1A). Survival probability reached 0% for the high expression group at 7,500 days. In addition, patients with *CPT1A* mutation(s) had a lower survival rate (median survival time of 1,100 days) than those without mutation(s) in *CPT1A* (median survival time of 4,300 days) (Fig 1B). At 3,200 days, the survival probability reached 0% for the group with a *CPT1A* mutation.

### Correlations between obesity status and tumor SUV mean by menopausal status

Tumor SUV$_{mean}$ is insignificantly (approaching statistical significance) negatively correlated with obesity status in pre-menopausal breast cancer patients (Fig 2A). In post-menopausal

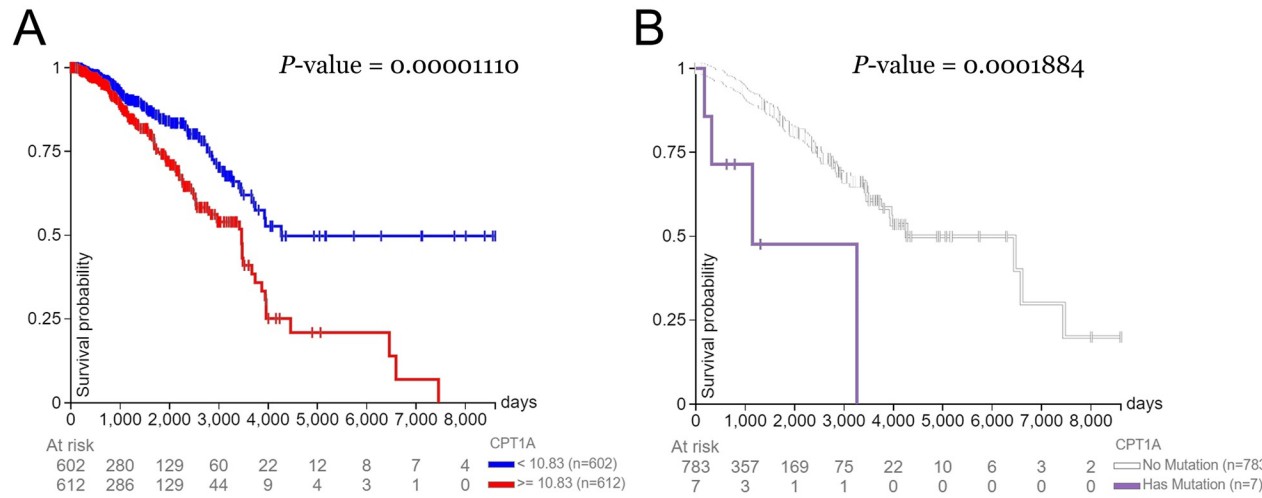

**Fig 1. Days after initial treatment vs. survival probability.** Correlations between days after initial treatment and survival probability in breast cancer patients **(A)** with high (> = 10.83 RKMP) vs. low (<10.83 RKMP) expression levels of *CPT1A* and **(B)** with or without mutations in *CPT1A*.

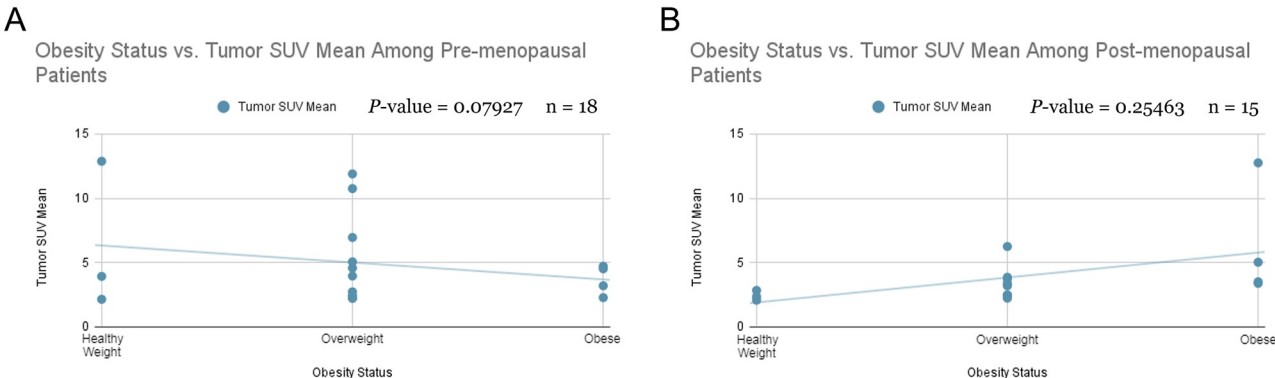

**Fig 2. Obesity status vs. lean body mass-corrected thymidine uptake.** Correlations between obesity status and lean body mass-corrected thymidine uptake value ($SUV_{mean}$) in breast cancer patients separated by **(A)** pre-menopausal status and **(B)** post-menopausal status. A two-sided Pearson correlation and Mann-Whitney test was performed, and the corresponding r value determined statistical significance. SUV—standardized uptake value.

breast cancer patients, tumor $SUV_{mean}$ is insignificantly positively correlated with obesity status (Fig 2B).

## $SUV_{mean}$ for tumor, adipose tissue, and organs by menopausal status

Tumor $SUV_{mean}$ in post-menopausal patients is significantly higher than tumor $SUV_{mean}$ in pre-menopausal patients (Fig 3). This same relationship is shown in heart $SUV_{mean}$, and the opposite relationship is shown in kidney $SUV_{mean}$.

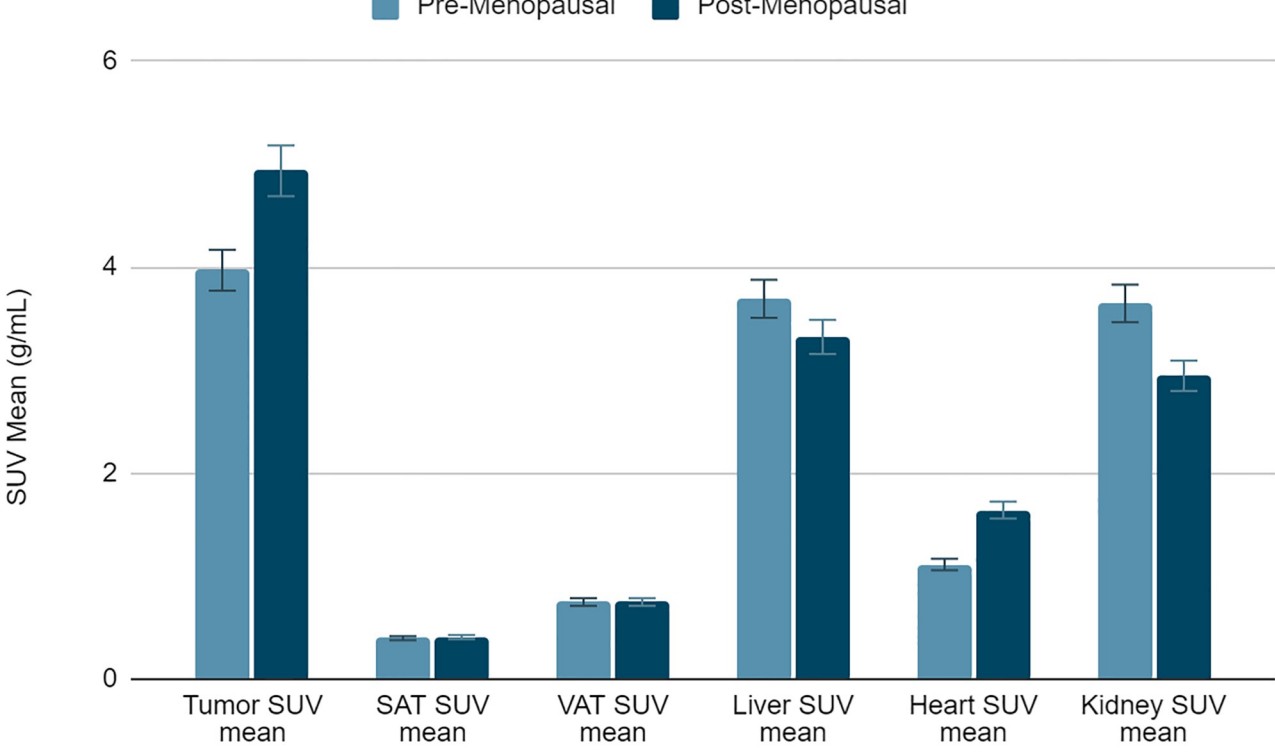

**Fig 3. Lean body mass-corrected mean standardized uptake value ($SUV_{mean}$) in tumor; subcutaneous adipose tissue (SAT); visceral adipose tissue (VAT) at the level of the S2 vertebrae; liver; heart; and kidney separated by menopausal status (n = 33).**

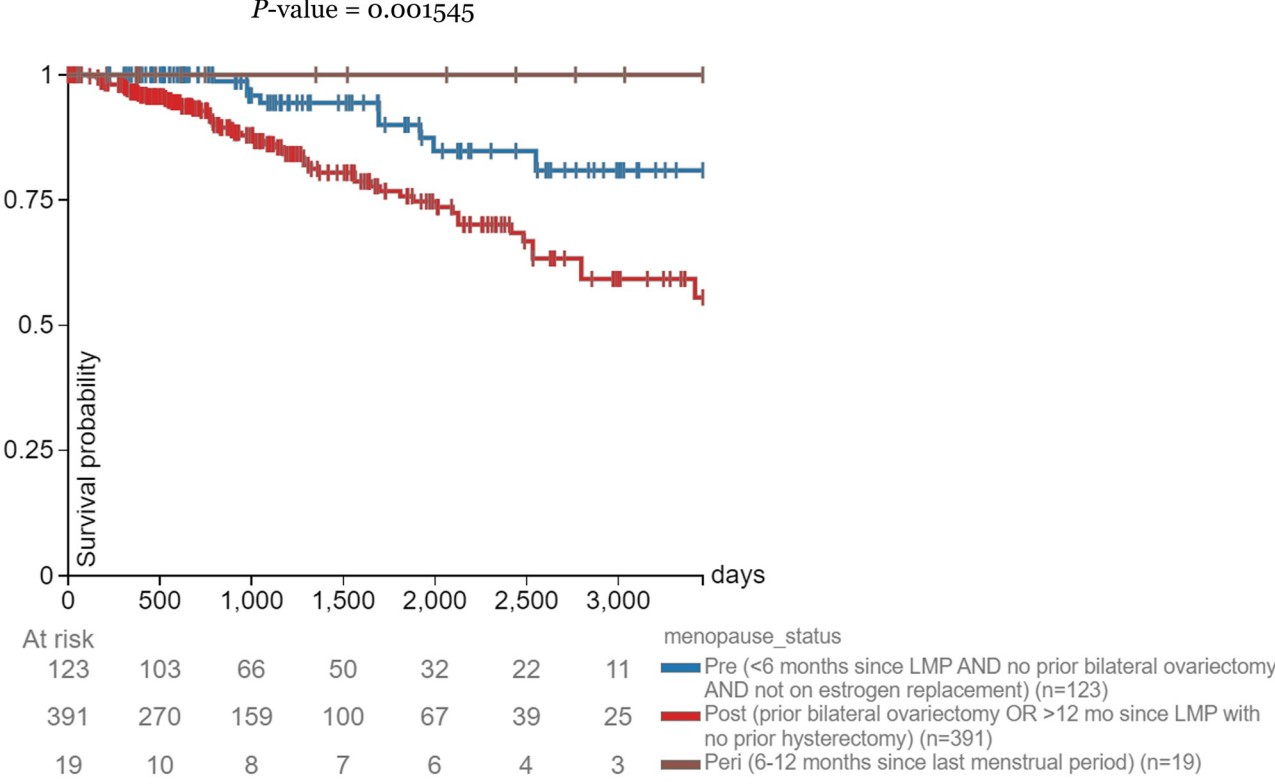

**Fig 4. Days after initial treatment vs. survival probability.** Correlations between days after initial treatment and survival probability in breast cancer patients with a high expression level of *CPT1A* ($>$ = 10.83 RKMP) separated by menopausal status.

### Survival probability for patients with high *CPT1A* expression by menopausal status

Pre-menopausal breast cancer patients with a high expression level of *CPT1A* had a higher survival rate than their post-menopausal counterparts (Fig 4). Peri-menopausal patients had a 100% survival probability throughout the study.

### Survival probability for patients with low *CPT1A* expression by menopausal status

Pre-menopausal breast cancer patients with a high expression level of *CPT1A* had a higher survival rate than their post-menopausal counterparts until the two groups' intersection at 3663 days (Fig 5). Peri-menopausal patients had the highest survival probability at 3663 days.

### Human expression of *CPT1A* by tissue type

Breast tissue had the fourth-highest median expression of *CPT1A*, preceded by organs of the gastrointestinal tract, including the colon, small intestine, and stomach, which are all adipose tissue-enriched microenvironments (Fig 6). Breast tissue was followed by kidney and liver tissue. Brain, blood, and pancreas tissue showed insignificant differences in *CPT1A* expression.

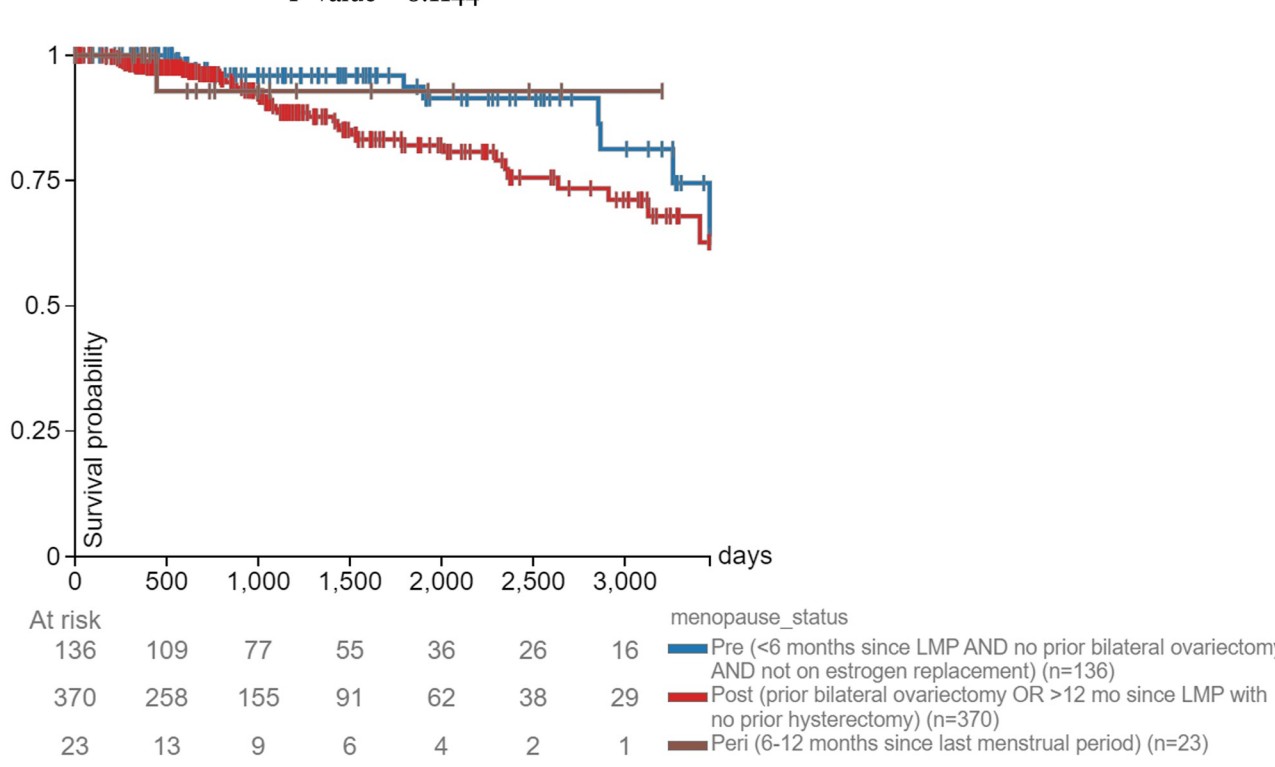

**Fig 5. Days after initial treatment vs. survival probability.** Correlations between days after initial treatment and survival probability in breast cancer patients with a low expression level of *CPT1A* (<10.83 RKMP) separated by menopausal status.

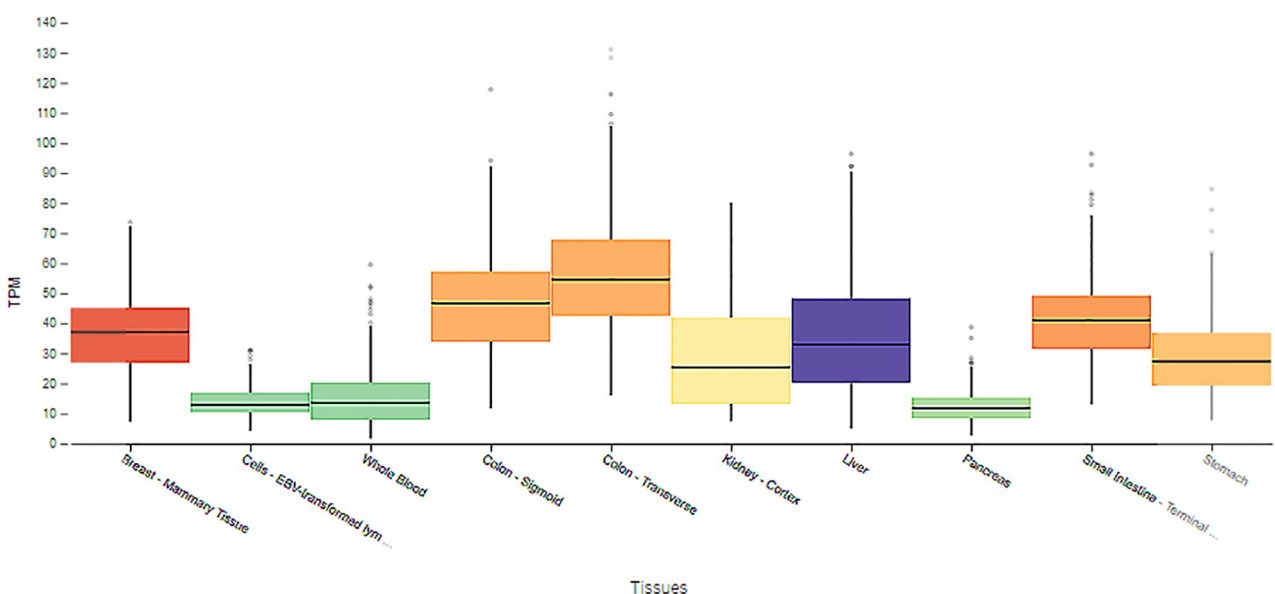

**Fig 6. Transcripts per million (TPM) of *CPT1A* separated by tissue types in which *CPT1A* is present.**

## Discussion

*CPT1A* is upregulated in breast cancer, worsening prognoses for high-risk patients. Although the role of obesity and menopausal status on breast cancer outcome has been studied before, we aimed to examine the lesser-known relationship between fatty acid metabolism and breast cancer outcome in the context of obesity and menopausal status. This study analyzes breast cancer patients in general, regardless of subtype, as there was limited data on subtype in both the image analysis and RNA sequencing visualization databases. *CPT1A* is upregulated in breast cancer, and inhibiting *CPT1A* activates cell apoptosis and suppresses cancer cell invasion [1]. Further, *CPT1A* is overexpressed in breast cancer tissues compared to normal breast tissues. This relationship is most strongly demonstrated in triple-negative breast cancer. We also observe an association between *CPT1A* expression and lymph node status, tumor size, tumor burden, histological grading, human epidermal growth factor receptor 2 (HER2) status, and molecular subtype. Fig 1A confirms these results and implies that 10.83 FPKM is the determining *CPT1A* expression level that produces significant differences in survival rate. Although previous studies have concluded that overexpression of *CPT1A* leads to poor breast cancer prognosis, to our knowledge no study has defined the deciding expression level: the expression level in which levels above or below it are predicted to result in significant differences in survival.

Fig 1B is consistent with the idea that *CPT1A* drives breast cancer metabolism [1]. A prominent clinical feature of *CPT1A* mutations in humans is elevated free carnitine levels [16], increasing the rate of fatty acid metabolism and energy for cancer cells to use and increasing proliferation and mortality. The connection between *CPT1A* mutations and breast cancer outcome is a gap in the literature, and this study explores the trend and the possible mechanisms behind it.

The image analysis portion of this study, represented in Fig 2A, strongly tended to confirm the paradoxical relationship between obesity and breast cancer in pre-menopausal women [12]. Increased serum estrogen levels are associated with poor breast cancer prognosis. In premenopausal women, estrogen is produced in the ovaries, and the low estradiol levels in patients are caused by negative feedback in the hypothalamic-pituitary axis suppressing ovarian function. On the other hand, our data from Fig 2B do not show a significant relationship between obesity and breast cancer in post-menopausal women. Estrogen synthesis in postmenopausal women occurs in adipose tissue, elevating estradiol levels in obese women compared to non-obese women. Although this relationship has been studied before, instead of using BMI as a continuous variable, this study uses obesity status, defined by BMI, as a categorical variable. BMI can reveal how height and weight affect breast cancer, but obesity status more explicitly represents how obesity affects outcome [17]. The analysis producing Fig 4 was not separated by obesity status, but post-menopausal patients still had a higher $SUV_{mean}$ than pre-menopausal patients. This served as a control for the image analysis data and showed that the bias towards post-menopausal patients should be further investigated. The comparably small sample size in the image analysis was limited by the number of eligible $^{18}$F-FLT scans on TCIA. The results approached statistical significance but, likely due to a type 2 error in the setting of limited available data, did not reach significance and would benefit from future followup studies with more samples. It should also be noted that our data on the interaction of menopausal status and BMI on breast cancer prognosis cannot distinguish between effects of age and of hormones *per se*. As aging is associated with increased ectopic lipid accumulation and increased *de novo* lipogenesis even independent of body weight, the effect of age could certainly confound the findings of the current study [18, 19]. However, due to the limited data available, we were not able to assess the impact of age as a continuous variable on the interaction between BMI, *CPT1*, and breast cancer outcomes.

To corroborate the image analysis results, we further explore the relationship between *CPT1A* and breast cancer, through the lens of the relationship between *CPT1A* and obesity. Past studies have found that *CPT1A* expression is positively correlated with BMI in humans [20]. Using RNA sequencing visualization, we filtered patients with high expression of *CPT1A*, using 10.83 FPKM as the determining level again, and explored the survival probabilities of patients in each menopausal stage within the expression level range, as portrayed by Fig 4. This study adds a new perspective to an established trend by breaking down the analysis of *CPT1A* and breast cancer by obesity and menopausal status. Interestingly, peri-menopausal patients with high *CPT1A* expression had a 100% survival probability across the study. This could be due to the small sample size in the peri-menopausal group (19 total), but it should be further explored. Fig 5 served as the control and increased confidence in the high *CPT1A* expression group data.

*CPT1A* is found in the liver, pancreas, kidney, brain, blood, and embryonic tissues [5]. The results of Fig 6 are partially expected because adipocytes are the predominant cell population in the breast, where they can secrete significant amounts of fatty acids as metabolic substrates for tumors [21]. Therefore, high CPT1A expression, which potentially leads to the high absorption of those fatty acids by the tumor, will take advantage of those fatty acid mobilizations, promoting increased cancer proliferation. Obesity, results in increased breast cancer incidence and breast tumor size, leading to an increased rate of metastasis formation and elevated mortality [2]. This effect potentially increases the number of fatty acid mobilizations for breast cancer cells to utilize. Previous studies have established that *CPT1A* is expressed in breast tissue, but this study compares this expression to other tissue types in which *CPT1A* is also highly expressed. Breast tissue is revealed as the tissue type with the highest *CPT1A* expression level, a feature not previously identified.

## Conclusion

In summary, fatty acid metabolism with respect to *CPT1A* was detrimental to breast cancer outcomes. We separated analyses by obesity and menopausal status, confirming the existing literature and presenting a new angle of breast cancer metabolism. PET and CT images were acquired through TCIA and analyzed in Fiji ImageJ, while RNA sequencing data was visualized in the Xena Functional Genomics Explorer, MARRVEL, and GTEx Portal. Although limited by the number of eligible $^{18}$F-FLT scans, we highlighted the importance of fatty acid metabolism in the study of breast cancer metabolism. Prospective studies should break down these analyses using cancer subtype, operative status, and more samples to better portray the effect of dysregulated fatty acid metabolism on breast cancer outcomes.

## Acknowledgments

We are grateful to Dr. Gang Peng in the Yale Cancer Center for his helpful input on the statistical analyses performed here.

## Author Contributions

**Conceptualization:** Ryan Liu.

**Data curation:** Ryan Liu.

**Formal analysis:** Ryan Liu.

**Funding acquisition:** Ryan Liu, Rachel J. Perry.

**Investigation:** Ryan Liu, Shyryn Ospanova.

**Supervision:** Rachel J. Perry.

**Writing – original draft:** Ryan Liu.

**Writing – review & editing:** Rachel J. Perry.

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
