## [Decision Letter · Decision Letter 0]

22 Nov 2022

PONE-D-22-26128The Impact of Variance in Carnitine Palmitoyltransferase-1 Expression on Breast Cancer Prognosis is Stratified by Clinical and Anthropometric FactorsPLOS ONE

Dear Dr. Perry,

Thank you for submitting your manuscript to PLOS ONE. After careful consideration, we feel that it has merit but does not fully meet PLOS ONE’s publication criteria as it currently stands. Therefore, we invite you to submit a revised version of the manuscript that addresses the points raised during the review process.

Specifically, the submission was evaluated by two reviewers, both of whom raised comments. Please have the concerns addressed point-by-point. We note that one reviewer has recommended that you cite specific previously published works. As always, we recommend that you please review and evaluate the requested works to determine whether they are relevant and should be cited. It is not a requirement to cite these works. 

We look forward to receiving your revised manuscript.

Kind regards,

Jianhong Zhou

Staff Editor

PLOS ONE

Journal Requirements:

“This study was supported in part by a Pilot Grant from the Yale Cancer Center, and by a dkNET Summer of Data student fellowship. The funders had no role in study design, data collection and analysis, decision to publish, or preparation of the manuscript. The authors declare that no conflicts of interest exist.”

“This study was supported in part by a Pilot Grant from the Yale Cancer Center, and by a dkNET Summer of Data student fellowship. The funders had no role in study design, data collection and analysis, decision to publish, or preparation of the manuscript.”

4. Please ensure that you refer to Figure 3 in your text as, if accepted, production will need this reference to link the reader to the figure.

5. Please upload a new copy of Figure 6 as the detail is not clear. Please follow the link for more information: " ext-link-type="uri" xlink:type="simple">https://blogs.plos.org/plos/2019/06/looking-good-tips-for-creating-your-plos-figures-graphics/"
https://blogs.plos.org/plos/2019/06/looking-good-tips-for-creating-your-plos-figures-graphics/

Reviewers' comments:

Reviewer's Responses to Questions

**Comments to the Author**

1. Is the manuscript technically sound, and do the data support the conclusions?

Reviewer #1: Yes

Reviewer #2: Yes

2. Has the statistical analysis been performed appropriately and rigorously? 

Reviewer #1: Yes

Reviewer #2: Yes

3. Have the authors made all data underlying the findings in their manuscript fully available?

Reviewer #1: Yes

Reviewer #2: Yes

4. Is the manuscript presented in an intelligible fashion and written in standard English?

Reviewer #1: Yes

Reviewer #2: Yes

5. Review Comments to the Author

Reviewer #1: Ryan Liu and authors in the manuscript “The Impact of Variance in Carnitine Palmitoyltransferase-1 Expression on Breast Cancer Prognosis is Stratified by Clinical and Anthropometric Factors” addressed the connection between breast cancer prognosis and fatty acid metabolism in cancer patients. CPT1A transports long-chain fatty acid to mitochondria for beta-oxidation and gene expression of this enzyme in the breast cancer tissue was used as a measurement of fatty acid metabolism rate within the cancer. To study this relationship, authors worked with available public databases and archive such as RNA-seq data and 18F-fluorothymidine PET-CT images and analyzed the data in relation to obesity and menopausal status using appropriate statistical methods. It is very interesting study, and their findings have significant impacts in the cancer metabolism field. In addition, the novelty of work is that authors skillfully combined research tools such as Fiji ImageJ, Xena Functional Genomics Explorer, MARRVEL, and GTEx Portal.

Despite the above, there are some major points that I would like to point out:

1. 10.83 FPKM is used as cut off level for CPT1A gene expression. How did authors determine this cut off level? It was not explained in the Method section.

2. Figures need to be referred in the Results, not in the Discussion.

3. In the sentence, the results on Figure 1A seems to be described in opposite way than the actual data.

4. Figure 4 on Page 6 and Figure 5 on Page 7 need to be described better and correct way. For these data set, could age rather than menopausal status play a role?

5. In Figure 6, lower CPT1A expression in brain parts is not surprising, because CPT1C is the major isoform in brain. However, the highest expression of CPT1A is observed in GI-tract (colon, duodenum, and small intestine). Can authors add these tissue in the Figure 6 in other to compare with breast tissue?

6. On page 9, the statement “these results are consistent with the fact that CPT1A is a major regulator of breast cancer” needs to be re-written. To make this statement, CPT1A activity assay or at least protein expression need to be determined. Authors for their interpretation, used only RNA-seq data and these data or actual gene expression of CPT1A is not confirmed by quantitative PCR in the tissue. Since metabolism in breast cancer cells not fully understood yet, a statement such as “a major regulator” should be avoided.

Minor points

1. On page 2, reference 3 was cited as “we demonstrated…” However, these authors were not listed as authors in the #3 manuscript. Please clarify this reference.

2. On page 2, “understanding” should be replaced by “understand”.

3. In the Figure 2, should it be simpler and easier to understand if words “normal healthy, overweight and obese” included instead of 2,3,4 numbers?

4. Number of participants (n) should be added in the legends for Figure 2 and 3.

5. Tissue names in Figure 6 on Page 8 are too small and not readable. Can authors replace this with higher resolution figure?

Reviewer #2: The authors analyzed the role of CPT1A differentially expression on Breast Cancer Prognosis concrning Clinical and Anthropometric Factors. The method is approperiatly designed and the manuscript is well presented. The results were validated using different measurement including CT scan and RNA-seq analysis. The work may gain so many interests, however, I have minor concerns:

- The figures are fuzzy. vector-based or high quality figures are required for the final version.

- The authors may refer to recent breast cancer progression works in the literature such as PMID: 35205681 and PMID: 34997055.

6. PLOS authors have the option to publish the peer review history of their article (what does this mean?). If published, this will include your full peer review and any attached files.

Reviewer #1: No

Reviewer #2: No

---

## [Author Response · Author response to Decision Letter 0]

13 Dec 2022

We thank the editors and reviewers for their time spent on this manuscript. Below please find our responses to all comments, queries, and concerns.

Journal comments:

Thank you for providing the links to the style templates. We have followed these requirements carefully.

“This study was supported in part by a Pilot Grant from the Yale Cancer Center, and by a dkNET Summer of Data student fellowship. The funders had no role in study design, data collection and analysis, decision to publish, or preparation of the manuscript. The authors declare that no conflicts of interest exist.”

“This study was supported in part by a Pilot Grant from the Yale Cancer Center, and by a dkNET Summer of Data student fellowship. The funders had no role in study design, data collection and analysis, decision to publish, or preparation of the manuscript.”

The funding statement is correct as written. We have now removed any funding-related text from the manuscript.

We have now provided links to the original data sets in the Materials and Methods section as well as in the Data Availability statement.

4. Please ensure that you refer to Figure 3 in your text as, if accepted, production will need this reference to link the reader to the figure.

In the revised version, we have ensured that all figures are referenced in the Results section.

5. Please upload a new copy of Figure 6 as the detail is not clear. Please follow the link for more information: https://blogs.plos.org/plos/2019/06/looking-good-tips-for-creating-your-plos-figures-graphics/" https://blogs.plos.org/plos/2019/06/looking-good-tips-for-creating-your-plos-figures-graphics/

A new copy of Figure 6 has now been uploaded. We hope that it will now meet the technical requirements of the journal. 

The reference list has been reviewed to ensure that it is complete and correct.

Reviewer comments:

Reviewer 1: Ryan Liu and authors in the manuscript “The Impact of Variance in Carnitine Palmitoyltransferase-1 Expression on Breast Cancer Prognosis is Stratified by Clinical and Anthropometric Factors” addressed the connection between breast cancer prognosis and fatty acid metabolism in cancer patients. CPT1A transports long-chain fatty acid to mitochondria for beta-oxidation and gene expression of this enzyme in the breast cancer tissue was used as a measurement of fatty acid metabolism rate within the cancer. To study this relationship, authors worked with available public databases and archive such as RNA-seq data and 18F-fluorothymidine PET-CT images and analyzed the data in relation to obesity and menopausal status using appropriate statistical methods. It is very interesting study, and their findings have significant impacts in the cancer metabolism field. In addition, the novelty of work is that authors skillfully combined research tools such as Fiji ImageJ, Xena Functional Genomics Explorer, MARRVEL, and GTEx Portal.

We thank the reviewer for considering this a “very interesting study,” and asserting that “the novelty of [this] work is that [we] skillfully combined research tools such as Fiji ImageJ, Xena Functional Genomics Explorer, MARRVEL, and GTEx Portal.” We are delighted that this reviewer considers this a “very interesting study, and [our] findings have significant impacts in the cancer metabolism field.”

Specific Points

1. 10.83 FPKM is used as cut off level for CPT1A gene expression. How did authors determine this cut off level? It was not explained in the Method section.

The reviewer makes a great point, and the manuscript added the explanation: “10.83 FPKM was the cutoff level for CPT1A gene expression, as the samples were divided at the median.” The protocol in UCSC Xena determined this.

2. Figures need to be referred in the Results, not in the Discussion.

This helpfully suggested change has been made. We have removed all figure descriptions from the Discussion and only reference figures when necessary to convey the point expanded upon in the Discussion.

3. In the sentence, the results on Figure 1A seems to be described in opposite way than the actual data.

We appreciate the reviewer for pointing out this mistake. It has been corrected: “Patients with a high expression level of CPT1A had a lower survival rate (medial survival time of 3,500 days) than those with low expression levels of CPT1A (medial survival time of 4,200 days).”

4. Figure 4 on Page 6 and Figure 5 on Page 7 need to be described better and correct way. For these data set, could age rather than menopausal status play a role?

We thank the reviewer for pointing out these errors. They have been corrected: “Pre-menopausal breast cancer patients with a high expression level of CPT1A had a higher survival rate than their post-menopausal counterparts” and “Pre-menopausal breast cancer patients with a high expression level of CPT1A observed a higher survival rate than their post-menopausal counterparts until the two group’s intersection at 3663 days.”

 We agree with the reviewer that age could play a role in the outcomes in this data set. We are not able to specifically test this question because of limited available data, but have added a comment on this important point in the Discussion:

“It should also be noted that our data on the interaction of menopausal status and BMI on breast cancer prognosis cannot distinguish between effects of age and of hormones per se. As aging is associated with increased ectopic lipid accumulation and increased de novo lipogenesis even independent of body weight, the effect of age could certainly confound the findings of the current study. However, due to the limited data available, we were not able to assess the impact of age as a continuous variable on the interaction between BMI, CPT1, and breast cancer outcomes.”

5. In Figure 6, lower CPT1A expression in brain parts is not surprising, because CPT1C is the major isoform in brain. However, the highest expression of CPT1A is observed in GI-tract (colon, duodenum, and small intestine). Can authors add these tissue in the Figure 6 in other to compare with breast tissue?

This is an important point that has been added in Figure 6. Breast tissue expression of CPT1A is now compared with blood, colon, kidney, liver, pancreas, small intestine, and stomach tissue.

6. On page 9, the statement “these results are consistent with the fact that CPT1A is a major regulator of breast cancer” needs to be re-written. To make this statement, CPT1A activity assay or at least protein expression need to be determined. Authors for their interpretation, used only RNA-seq data and these data or actual gene expression of CPT1A is not confirmed by quantitative PCR in the tissue. Since metabolism in breast cancer cells not fully understood yet, a statement such as “a major regulator” should be avoided.

 The reviewer brings up a valid point. We have replaced the claim with a less definitive statement: “Figure 1b is consistent with the idea that CPT1A drives breast cancer metabolism.”

7. On page 2, reference 3 was cited as “we demonstrated…” However, these authors were not listed as authors in the #3 manuscript. Please clarify this reference.

 This helpful suggestion has been incorporated, by removing “we demonstrated” and replacing it with “as demonstrated in the literature.”

8. On page 2, “understanding” should be replaced by “understand”.

This helpful correction has been made.

9. In the Figure 2, should it be simpler and easier to understand if words “normal healthy, overweight and obese” included instead of 2,3,4 numbers?

 The reviewer makes a great point. Figure 2 now has qualitative x-axis labels.

10. Number of participants (n) should be added in the legends for Figure 2 and 3.

This helpful suggestion has been incorporated.

11. Tissue names in Figure 6 on Page 8 are too small and not readable. Can authors replace this with higher resolution figure?

 The figures have been replaced with their high-resolution counterparts.

Reviewer 2: The authors analyzed the role of CPT1A differentially expression on Breast Cancer Prognosis concrning Clinical and Anthropometric Factors. The method is approperiatly designed and the manuscript is well presented. The results were validated using different measurement including CT scan and RNA-seq analysis.

We thank Reviewer 2 for stating that “the method is appropriately designed and the manuscript is well presented.” We are pleased that the reviewer values the validation of the conclusions.

Specific Points

1. The figures are fuzzy. vector-based or high quality figures are required for the final version.

The figures have been replaced with their high-resolution counterparts.

2. The authors may refer to recent breast cancer progression works in the literature such as PMID: 35205681 and PMID: 34997055.

 We have reviewed the useful literature mentioned, and thank the reviewer for the suggestion. These papers have now been cited in the Introduction.

---

## [Decision Letter · Decision Letter 1]

19 Jan 2023

The Impact of Variance in Carnitine Palmitoyltransferase-1 Expression on Breast Cancer Prognosis is Stratified by Clinical and Anthropometric Factors

PONE-D-22-26128R1

Dear Dr. Perry,

We’re pleased to inform you that your manuscript has been judged scientifically suitable for publication and will be formally accepted for publication once it meets all outstanding technical requirements.

Kind regards,

Divyansh Agarwal

Academic Editor

PLOS ONE

Additional Editor Comments (optional):

Reviewers' comments:

Reviewer's Responses to Questions

**Comments to the Author**

1. If the authors have adequately addressed your comments raised in a previous round of review and you feel that this manuscript is now acceptable for publication, you may indicate that here to bypass the “Comments to the Author” section, enter your conflict of interest statement in the “Confidential to Editor” section, and submit your "Accept" recommendation.

Reviewer #1: All comments have been addressed

Reviewer #2: All comments have been addressed

2. Is the manuscript technically sound, and do the data support the conclusions?

Reviewer #1: Yes

Reviewer #2: Yes

3. Has the statistical analysis been performed appropriately and rigorously? 

Reviewer #1: Yes

Reviewer #2: Yes

4. Have the authors made all data underlying the findings in their manuscript fully available?

Reviewer #1: Yes

Reviewer #2: Yes

5. Is the manuscript presented in an intelligible fashion and written in standard English?

Reviewer #1: Yes

Reviewer #2: Yes

6. Review Comments to the Author

Reviewer #1: Ryan Liu and authors have addressed most of the comments in the revision of their manuscript “The Impact of Variance in Carnitine Palmitoyltransferase-1 Expression on Breast Cancer Prognosis is Stratified by Clinical and Anthropometric Factors”.

Though, a few more minor concerns I would like to point out:

1. In the Discussion, instead of citing just Figure 1b, I like the actual sentence that has been removed in the revised version. In continuation of the sentence, how about this idea: “These results are consistent with the idea that breast cancer metabolism could be (might be?) more dependent on CPT1a-mediated mitochondrial fatty acid oxidation”?

2. In the end of the Discussion, the last sentence “Breast tissue is revealed as the tissue type with the highest CPT1A expression level, a feature not previously identified”, are the authors talking about normal breast tissue or breast cancer tissue? Since Figure 6 is so blurry, I really cannot read that 3 words. At least, based on the updated version of Figure 6, they should say “one of the highest…”.

3. Does number of participants n=33 in the legend for Figure 3 mean 33 for each groups or total for 2 groups? It needs to be clarified.

4. Tissue names in Figure 6 are still blurry and difficult to read. Can authors include tissue names in the figure legend (for example name them in order from left to right)?

Reviewer #2: I think the reviewers have done fair efforts addressing reviewers comments. I was wish for more literature. but overall 21 papers in the reference is fine.

7. PLOS authors have the option to publish the peer review history of their article (what does this mean?). If published, this will include your full peer review and any attached files.

Reviewer #1: No

Reviewer #2: **Yes: **ABEDALRHMAN ALKHATEEB

---

## [Editor Report · Acceptance letter]

23 Jan 2023

PONE-D-22-26128R1 

The Impact of Variance in Carnitine Palmitoyltransferase-1 Expression on Breast Cancer Prognosis is Stratified by Clinical and Anthropometric Factors 

Dear Dr. Perry:

I'm pleased to inform you that your manuscript has been deemed suitable for publication in PLOS ONE. Congratulations! Your manuscript is now with our production department. 

Kind regards, 

on behalf of

Dr. Divyansh Agarwal 

Academic Editor

PLOS ONE